# Hypertrophic Pyloric Stenosis in an Adolescent Girl: An Atypical Presentation of an Unexpected Disease

**DOI:** 10.3390/diseases11010019

**Published:** 2023-01-28

**Authors:** Simona Gatti, Francesca Piloni, Edoardo Bindi, Alba Cruccetti, Carlo Catassi, Giovanni Cobellis

**Affiliations:** 1Department of Pediatrics, Università Politecnica delle Marche, Via Corridoni 11, 60123 Ancona, Italy; 2Department of Pediatric Surgery, Università Politecnica delle Marche, 60123 Ancona, Italy

**Keywords:** hypertrophic pyloric stenosis, adolescent, late presentation, coffee-ground emesis, melena

## Abstract

Hypertrophic pyloric stenosis is a common cause of vomiting in the first few weeks of life, but in rare cases, it may occur in older subjects with a major risk of delayed diagnosis and complications. We describe the case of a 12-year-and-8-month-old girl who presented to our department for epigastric pain, coffee-ground emesis, and melena, which arose after taking ketoprofen. An abdomen ultrasound showed thickening (1 cm) of the gastric pyloric antrum, while upper-GI endoscopy documented esophagitis and antral gastritis with a non-bleeding pyloric ulcer. During her hospital stay, she had no further episodes of vomiting and was therefore discharged with a diagnosis of “NSAIDs-induced acute upper gastrointestinal tract bleeding”. After 14 days, following recurrence of abdominal pain and vomiting, she was hospitalized again. At endoscopy, pyloric sub-stenosis was found, abdominal CT showed thickening of large gastric curvature and pyloric walls, and an Rx barium study documented delayed gastric emptying. On suspicion of idiopathic hypertrophic pyloric stenosis, she underwent Heineke–Mikulicz pyloroplasty with resolution of symptoms and restoration of a regular caliber of the pylorus. Hypertrophic pyloric stenosis, although occurring rarely in older children, should be taken into account in the differential diagnosis of recurrent vomiting at any age.

## 1. Introduction

Hypertrophic pyloric stenosis (HPS) is a common condition in the neonatal period (incidence of 1.5–3 per 1000 live births) [1] with an unclear etiology [2]. Few cases have been described in adults, with fewer than 300 cases reported in the literature [3,4,5], while presentation in adolescence is definitely the rarest, with only 3 cases described so far [6,7,8]. In an older child/adolescent the diagnostic approach requires a high index of suspicion and exclusion of other conditions leading to gastric outlet obstruction (GOO). The work-up is based on imaging studies and upper gastrointestinal endoscopy. Definitive treatment includes endoscopic or surgical procedures.

We report the case of an adolescent girl that, after an extensive gastroenterological and surgical work-up, was successfully treated for late onset HPS.

Case description A 12-year-and-8-month-old girl accessed the emergency department for a recent history of epigastric pain associated with multiple episodes of coffee-ground emesis and melena, which arose after taking two doses of ketoprofen for gingivitis. She was the second child of non-consanguineous parents, born at term from vaginal delivery (birth weight: 2810 kg). She had an unremarkable medical history, except for a bilateral inguinal hernia surgically corrected at three years of age. There was no history of cow’s milk or other allergies and no exposure to long-term medications or to active or passive smoking. Initial laboratory exams documented a normal full blood count (hemoglobin levels 13.2 g/dL) and coagulation pattern within normal limits. On abdominal ultrasound, a “marked parietal thickening (approximately 1 cm) of the gastric pyloric antrum supported by submucosal and mucosal edema” was described (Figure 1).

The patient was therefore admitted to our Pediatric Department where, upon physical evaluation, she presented good general conditions; her blood pressure was 110/59 mmHg, and the heart rate was 110 bpm. Weight and height were both on the 25° centile for age. At a careful medical recall, she reported a history of intermittent nausea and precocious satiety in the previous year. On admission, she was started on IV fluids and gastroprotective therapy, continuous monitoring of vital parameters, and, after surgical evaluation, she underwent urgent upper GI-endoscopy 2 h after admission. Endoscopy showed distal esophagitis, and abundant coffee-ground fluid (700 mL) was found and aspirated from the stomach. The antral mucosa had a nodular aspect, and a pre-pyloric, non-bleeding ulcer with fibrin-covered fundus was documented. A narrow pyloric channel was noted, but the pediatric gastroscope could pass through (Figure 2).

Histologic examination documented hyperplastic-reparative aspects of the distal esophagus mucosa and reactive changes in the gastric body and antral mucosa. Standard stains and immune-histochemistry were negative for CMV and Helicobacter pylori. Celiac serology and fecal calprotectin were negative. During hospitalization, the patient had no further episodes of vomiting and maintained good general clinical conditions and stable vital parameters. The clinical picture was considered compatible with a diagnosis of “NSAIDs-induced acute upper gastrointestinal tract bleeding” and the girl was discharged on oral proton-pump inhibitors. Ten days after discharge, she reported loss of appetite and precocious satiety, which was associated with weight loss of about 1 kg. Three days after, she experienced episodes of abdominal pain and nausea after a large meal, followed by non-bilious vomiting. For the subsequent recurrence of two episodes of profuse and coffee-ground emesis in the following days, she was readmitted in the hospital. At a second endoscopy, a fibrin-covered pyloric ulcer (approximately 8 mm in size and not actively bleeding) and a substenotic appearance of the pylorus were found (the neonatal endoscope was necessary to intubate the pyloric channel). (Figure 3)

A contrast barium Rx study documented delayed gastric emptying with slow passage of fluids through the pylorus. In consideration of the pyloric substenosis, an abdominal CT scan was performed, which excluded neoplastic lesions but confirmed the presence of diffuse wall thickening of the large gastric curvature and pylorus (Figure 4).

On suspicion of idiopathic hypertrophic pyloric stenosis, the patient was transferred to the Department of Pediatric Surgery, where she was maintained on total parenteral nutrition. On day 22 from admission, she underwent Heineke–Mikulicz pyloroplasty (HMP) by an open approach. At laparotomic exploration, the pylorus appeared thickened and hypertrophic. Full-thickness biopsies of the pyloric wall were performed and documented foveolar hyperplasia, mild inflammatory lymphocytic infiltrate of the lamina propria, and blood congestion of the submucosa; it also confirmed pyloric muscle hypertrophy. A pyloroplasty was then performed by the technique of Heineke–Mikulicz via a full-thickness longitudinal incision of the muscular wall of the pylorus, which was then sutured transversely. After pyloroplasty, water was infused via a nasogastric tube, which showed patency of the pylorus and absence of perforation. In the postoperative period, the patient gradually was restarted on oral diet, initially with creamy and later with solid foods, with no occurrence of vomiting or other gastrointestinal symptoms. Ultrasound examination at 30 days post-op was within normal limits (Figure 5).

At endoscopy performed three months after surgery, a pylorus of regular caliber and normal gastric and duodenal mucosa were documented. At the nine-month medical follow-up, the girl did not complain of any symptom and had regained 2 kg of weight.

## 2. Discussion

We described the case of a late-onset pyloric hypertrophy in an adolescent Italian girl and compared this with similar cases from the literature.

HPS is the principal cause of GOO in the first months of life (typically presenting within the first 12 weeks of life with a higher prevalence in males). At this age, incidence is of 1.5–3 per 1000 live births [1], and the etiology is not completely understood [2], with possible involvement of genetic, hormonal, and environmental factors. Beyond infancy, HPS is extremely rare, with other causes (both congenital and acquired) of GOO becoming prevalent in older children and adults. These can be grouped in three categories: 1. congenital anomalies (diaphragms and webs, atresia, or congenital microgastria), 2. secondary forms (peptic disease or severe inflammation such as in Crohn’s disease, chemical injury or foreign body ingestion, or neoplastic), and 3. primary acquired [9,10,11]. The latter is also known as Jodphur’s disease (from the name of the city where was originally described in India) [9], and it is characterized by the absence of pyloric hypertrophy and is probably related to a motility disturbance. Triggering factors have been postulated to contribute to the delayed onset of symptoms both in the pediatric and adult forms of hypertrophy of the pylorus including edema, pylorospasm, neuromuscular incoordination due to changes in Auerbach’s plexus, or inflammation [12,13].

Our extensive literature review identified a total of 21 cases of HPS described in children with onset of symptoms after three months of age (range: seven months–17 years) [6,7,8,14,15,16,17,18,19,20,21], with only three cases reported in adolescents (14, 15, and 17 years, respectively) [6,7,8] (Table 1).

In all the reported cases, vomiting was the presenting complaint, accompanied by weight loss or growth failure, abdominal pain, and precocious satiety. Apart from our case, gastrointestinal hemorrhage (hematemesis and/or melena) was reported only in another adolescent girl [8]. Interestingly, in both cases, gastritis was documented. The finding of a gastritis or peptic disease was restricted to older patients; it is therefore possible to postulate a relation between long-standing gastric stasis and a subsequent reactive inflammation. Besides our case, a Helicobacter pylori-associated pre-pyloric ulcer was described in a previous case report [7].

The peculiarity of our case was certainly the age and gender of the patient, the presentation with hematemesis, and the confounding initial endoscopic picture, which showed signs of inflammation compatible with NSAID’s gastritis. Recurrence of symptoms requiring a second endoscopic evaluation led to further investigations and finally allowed us to postulate a diagnosis of HPS.

In the management of suspected HPS, first-level investigations include abdominal ultrasound that can document an increased pylorus thickness (in infants: thickness > 3 mm in a transverse section [22,23] and in adults > 8 mm [24]). A subsequent step is to perform a contrast study both to rule out other conditions (malrotation, hiatal ernia) and to document the narrowing of the pyloric lumen with associated delayed gastric emptying and gastric enlargement. Abdominal computed tomography (CT) and/or magnetic resonance imaging (MRI) are essential to document mass lesions or ab estrinseco compressions and to definitively diagnose secondary pyloric stenosis. Upper gastrointestinal endoscopy is a fundamental step in the diagnostic algorithm of late-onset HPS for several reasons: 1. it allows a direct visualization and size estimation of the narrowed pyloric channel; 2. gastric and duodenal mucosa status can be assessed, and presence of mucosal diseases can be ruled out; 3. mucosal and submucosal biopsies can be performed; and 4. some endoscopic treatments can alleviate symptoms (balloon dilatation and/or cauterization).

Definitive diagnosis of HPS can be based on first-level imaging (ultrasound is sufficient in typical neonatal cases) or require further investigations as previously indicated. In our case, we performed both CT and a barium study, and two endoscopic assessments were needed before a definitive diagnosis was made. A histological confirmation (hypertrophy and hyperplasia of the inner circular muscle layer of the pylorus, some degree of fibrosis, and reactive changes in the mucosal layer) is not always necessary, and most of the time, it is not easy to obtain a full-thickness biopsy.

In terms of treatment, both endoscopic and surgical strategies have been reported in pediatric-onset HPS. We found endoscopic balloon dilatation attempted in eight cases [14,16,19,21], and in four of them, it was curative [16] with no requirement for surgical options.

The treatment of choice for neonatal HPS is pyloromyotomy, which was traditionally performed under laparotomy and subsequently adapted to laparoscopy [25,26,27]. More recently endoscopic piloromiotomy has also been described [28]. In pediatric cases of late-onset HPS, pyloromyotomy was performed as the first treatment in two cases [15,18], and in two other cases [8,21], this was done after failure of less invasive treatments. Resolution was achieved in all the cases.

Pyloroplasty, initially performed in 1886 by Dr. Heineke and subsequently by Dr. Mikulicz, relies on longitudinally opening the pylorus and transverse closure, and it is the most common technique for the treatment of peptic ulcer disease. The surgical principle on which this technique is based, given its effectiveness, has also been adopted for the treatment of other pathological stenoses, such as intestinal strictures secondary to Crohn’s disease. Given the patient’s age and the thickness of the pylorus, we decided to perform an HMP with an open approach in order to have safer control of the pylorus and to be able to perform a pyloroplasty that would be effective while minimizing the risk of recurrence or iatrogenic injury. In fact, although in our center the approach of choice for pyloroplasty in the infant is laparoscopic, in this case, we felt that the lower tactile feedback of laparoscopy might expose us to a higher risk of perforation or incomplete muscle opening. HMP was the technique of choice in three other cases reported in the literature [6,17,19] and, as in our experience, was resolutive in all of them.

A more invasive surgical technique was required in the remaining nine cases, including distal gastrectomy with Billroth-I or Roux-en-y reconstruction [7,14,16,20]. In three cases, this technique was performed after failure of minimally invasive procedures (endoscopic balloon dilatation, botulin injection, and laser coagulation) [7,14,16].

All the reported cases and our personal experience indicate a successful outcome of the surgical strategy with a complete restitution ad integrum of pyloric function and symptom resolution, regardless the specific surgical technique. In this setting, the choice of more aggressive surgical strategies is therefore questionable unless specific contra-indication to less invasive procedures exists or there is a specific necessity for diagnostic purposes [21] (e.g., to collect full-thickness biopsies).

## 3. Conclusions

HPS should be considered in the differential diagnosis of older children and adolescents with recurrent vomiting or hematemesis, even when other causes seem to be more plausible. A multistep work-up, including upper gastrointestinal endoscopy and abdominal CT scan, should be followed in order to reach the final diagnosis, and treatment by a minimally invasive endoscopic/surgical procedure should be preferred.

## Figures and Tables

**Figure 1 diseases-11-00019-f001:**
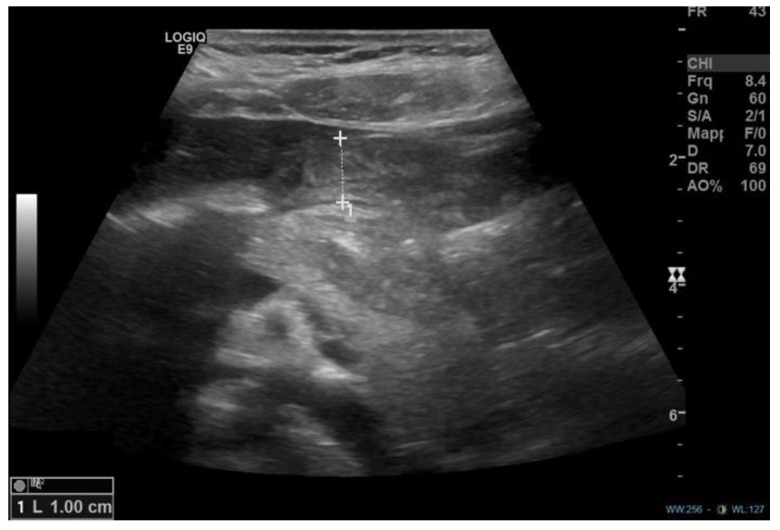
Abdominal ultrasound showing parietal thickening (1 cm) of the gastric pyloric antrum in the longitudinal plane.

**Figure 2 diseases-11-00019-f002:**
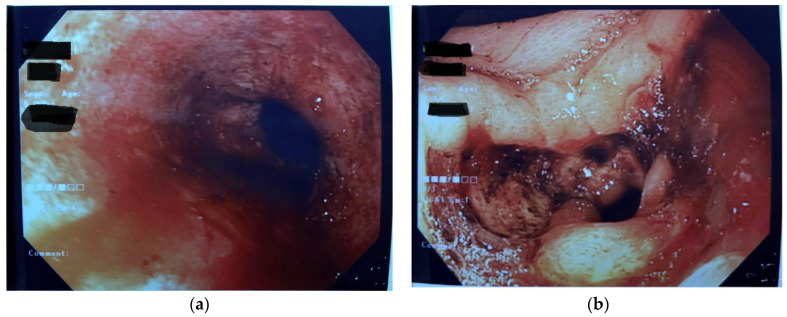
First endoscopy documenting esophagitis (**a**) and abundant coffee-ground fluid in the stomach and nodular antral gastritis (**b**).

**Figure 3 diseases-11-00019-f003:**
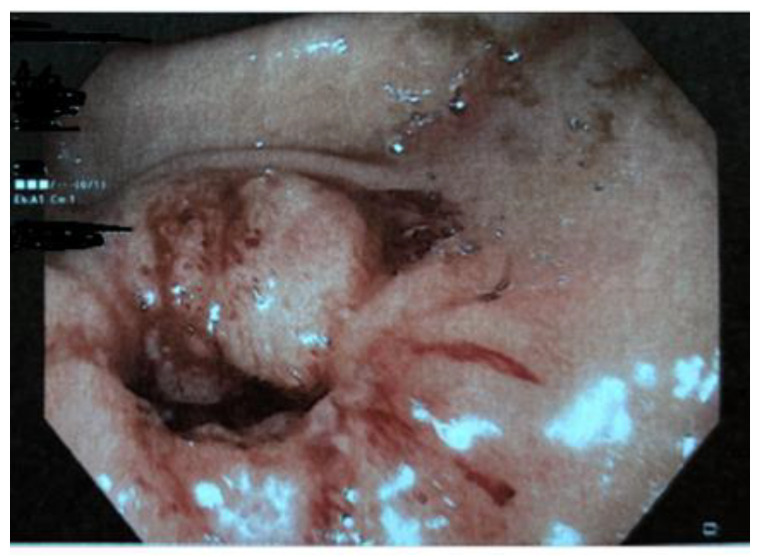
Second endoscopy documenting substenotic appearance of the pyloric channel.

**Figure 4 diseases-11-00019-f004:**
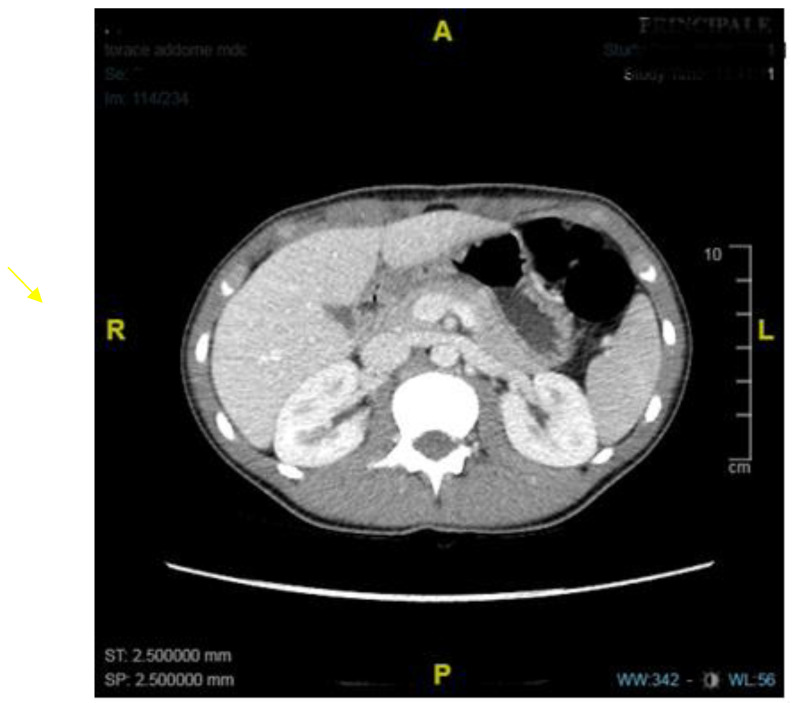
Abdominal CT demonstrating circumferential thickening of the pylorus and of the large gastric curvature in the transverse plane.

**Figure 5 diseases-11-00019-f005:**
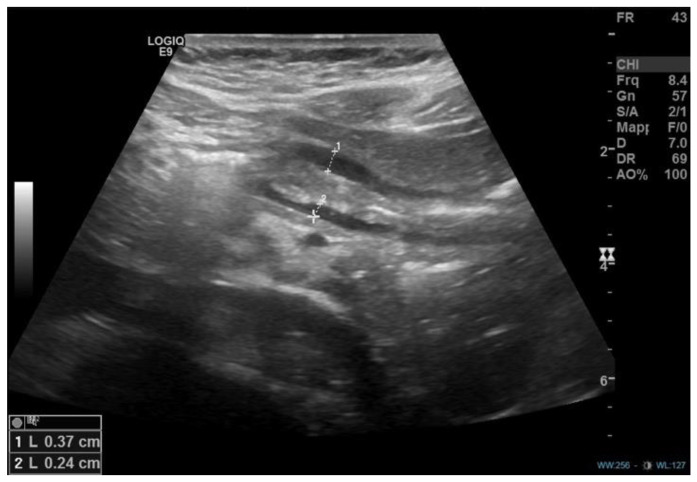
Post-operative abdominal ultrasound showing regular caliber of the pylorus.

**Table 1 diseases-11-00019-t001:** Clinical features and management of cases of late-onset (>3 months) HPS reported in children and adolescents.

Author, Year	Number of Cases	Age	Symptoms	Investigations (Prior to Surgery)	Further Endoscopic Findings	Treatments
Selzer et al. [6], 2009	1	14 years	Diarrhea, nausea, vomiting, abdominal pain	UGI series	Eosinophilic esophagitis	HMP
Mahalik et al. [15], 2010	1	4,5 years	Non bilious vomiting	US, UGI series, CT scan	-	OP
Boybeyi et al. [16], 2010	11	3,6 (2–8) years (mean age, range)	Vomiting (5 cases) Vomiting + abdominal pain (4 cases) Vomiting + weight loss (2 cases)	US + UGI series (6 cases), UGI series (5 cases), EGDS	Gastric edema and hyperemia (2 cases)	EBD (4 cases)–EBD + B-I (1 case)–B-I (6 cases)
Bajpai et al. [17], 2013	1	8 years	Non bilious vomiting and poor growth	US and X-Ray, EGDS, CT scan, UGI series	-	HMP + temporary jejunostomy
Parnall et al. [7], 2016	1	15 years	Vomiting, early satiety, failure to thrive	US, CT	Prepyloric ulcer (Hp positive)	Botulinum toxin injection (failure), distal gastrectomy with Billroth I reconstruction
Wolf et al. [8], 2016	1	17 years	Abdominal pain, bloating, early satiety, one episode of upper gastrointestinal hemorrage, anemia	EGDS (2), MRI enterography, gastric emptying study using Technetium-99	Severe gastritis	Botulinum injection (failure), LP
Al-Mayoofet al. [18], 2016	1	7 months	Vomiting, weight loss	UGI series, US	-	OP
Bartlett et al. [19], 2018	1	12 years	Vomiting, failure to thrive	EGDS, US.	Previous history of eosinophilic esophagitis. Gastritis.	EBD + Botulinum injection (failure) HMP + temporary gastrostomy
Oswari et al. [20], 2020	1	11 years	Vomiting, previous history of epigastric trauma	UGI series, EGDS, CT scan		B-I
Plessi et al. [14], 2021	1	12 years	Vomiting, growth failure (underlying diagnosis: Down’s Syndrome)	Abdominal US, UGI series (2), MRI, EGDS (2)	-	EBD + electrosurgical incisions (failure), distal gastrectomy with Roux-en-y reconstruction
Iacoviello et al. [21], 2022	1	3 years	Vomiting, rumination and weight loss	Abdominal US and X-Ray, UGI series (2), CT scan, EGDS, MRI.	-	EBD (failure) OP

US: ultrasound, CT: computed tomography, UGI: upper gastrointestinal contrast study, EGDS: esophago-gastroduodenal-endoscopy, MRI: magnetic resonance imaging, OP: open pyloromyotomy, LP: laparoscopic pyloromyotomy, HMP: Heineke–Mikulicz pyloroplasty, EBD: endoscopic balloon dilation, B-I: Billroth I.

## Data Availability

The data presented in this study are available on request from the corresponding author.

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
