# Peer review of "Hypertrophic Pyloric Stenosis in an Adolescent Girl: An Atypical Presentation of an Unexpected Disease"

_diseases, 2023, doi:10.3390/diseases11010019_

Round 1

Reviewer 1 Report

Gatti S et al. describe the case of an adolescent girl affected by hypertrophic pyloric stenosis which started clinically as a gastrointestinal bleeding.

The Case Report is interesting and well described.

The present reviewer has only minor concerns that should be addressed:

1) Some information regarding the clinical history of the patient, such as birth order and weight, type of delivery, age and smoking status of the mother, exposition to erythromicin,  presence of a cow's milk protein allergy, could be included in the case description if available;

2) Since this disease is about five-folds more frequent in males that females also in cases diagnosed later than 8-12 weeks of age, the role of the gender should be briefly discussed in the Discussion Section;

3) In the Conclusion Section, line 195, after "A multistep work-up" I should add the phrase "including upper gastrointestinal endoscopy and abdominal CT scan";

4) Line 65: istopathology should be histopathology.

Author Response

We thank the reviewers for their evaluation of our manuscript and their meaningful comments. We have modified the manuscript according to the comments of both the reviewers. Please find our point-by-point reply to the referee’s comments below.

# Reviewer 1

1) Some information regarding the clinical history of the patient, such as birth order and weight, type of delivery, age and smoking status of the mother, exposition to erythromicin,  presence of a cow's milk protein allergy, could be included in the case description if available;

We have added these informations in the case description session (lines 42-46)

2) Since this disease is about five-folds more frequent in males that females also in cases diagnosed later than 8-12 weeks of age, the role of the gender should be briefly discussed in the Discussion Section;

Thanks for the comment, we have now added few lines in the discussion section (lines 114 and 145)

3) In the Conclusion Section, line 195, after "A multistep work-up" I should add the phrase "including upper gastrointestinal endoscopy and abdominal CT scan";

We have added this in the final paragrah (lines 205-206).

4) Line 65: istopathology should be histopathology.

Thanks, we have now corrected the word istochemistry with histochemistry  (line 69)

Reviewer 2 Report

 I enjoyed reading the case report “HYPERTROPHIC PYLORIC STENOSIS IN AN ADOLESCENT GIRL: AN ATYPICAL PRESENTATION OF AN UNEXPECTED DISEASE”. The authors present a case of a 12 year old girl with pyloric hypertrophy.

I have a few questions:

Could auto-immune inflammation have been a differential diagnosis?

Would laparoscopic surgery have been an option instead of open surgery?

Why was chosen to perform a Heineke-Mikulicz pyloroplasty on day 22 instead of earlier?

The name Heineke-Mikulicz is mentioned twice which is confusing. Consider correcting this.

How long was the follow-up?

Author Response

We thank the reviewer for meaningful comments.

1)Could auto-immune inflammation have been a differential diagnosis?

We agree with the reviewer that some autoimmune conditions (particularly Crohn’s disease) could have been an underlying diagnosis. We ruled out Crohn’s disease, autoimmune gastritis and celiac disease based on the endoscopic and histological characteristics and on serological markers (we have added few line 70).

2)Would laparoscopic surgery have been an option instead of open surgery?

We agree that a laparoscopic approach could have been attempted, as we usually do in infants with the typical presentation of pyloric stenosis. In this case, instead, we decided to perform an open surgery in order to have a safer control of the region and to be able to perform a pyloroplasty. In fact, although in our center the approach of choice for pyloroplasty in the infant is laparoscopic, in this case we felt that the lower tactile feedback of laparoscopy might expose us to a higher risk of perforation or incomplete muscle opening (as stated in lines 186-89).

3)Why was chosen to perform a Heineke-Mikulicz pyloroplasty on day 22 instead of earlier?

We performed several investigations before reaching the suspicion of such a rare condition. Furthermore the girl required some days of parenteral nutrition in order to gain some weight before surgery.

4)The name Heineke-Mikulicz is mentioned twice which is confusing. Consider correcting this.

Thanks for the observation, we have now corrected this.

5)How long was the follow-up?

Last medical follow-up was at 9 months, we have specified this (line 108).